# Measuring the impact of nonpharmaceutical interventions on the SARS-CoV-2 pandemic at a city level: An agent-based computational modelling study of the City of Natal

**Paulo Henrique Lopes**[1,2☯], **Liam Wellacott**[3☯], **Leandro de Almeida**[4,7], **Lourdes Milagros Mendoza Villavicencio**[5], **André Luiz de Lucena Moreira**[1,2], **Dhiego Souto Andrade**[1,2], **Alyson Matheus de Carvalho Souza**[1,2], **Rislene Katia Ramos de Sousa**[6], **Priscila de Souza Silva**[6], **Luciana Lima**[6], **Michael Lones**[3], **José-Dias do Nascimento Jr.**[4], **Patricia A. Vargas**[3], **Renan Cipriano Moioli**[1,2,3], **Wilfredo Blanco Figuerola**[1,8], **César Rennó-Costa**[1,2,3]*

**1** Bioinformatics Multidisciplinary Environment of the Digital Metropolis Institute, Federal University of Rio Grande do Norte, Natal, Brazil, **2** Digital Metropolis Institute, Federal University of Rio Grande do Norte, Natal, Brazil, **3** Robotics Laboratory, Edinburgh Centre for Robotics, Heriot-Watt University, Edinburgh, United Kingdom, **4** Physics Department, Federal University of Rio Grande do Norte, Natal, Brazil, **5** Centro de Ciências Exatas e da Terra, Federal University of Rio Grande do Norte, Natal, Brazil, **6** Demography Graduate Program, Federal University of Rio Grande do Norte, Natal, Brazil, **7** Laboratório Nacional de Astrofísica, Itajubá, MG, Brazil, **8** Computer Science Department, State University of Rio Grande do Norte, Natal, Brazil

☯ These authors contributed equally to this work.
* cesar@imd.ufrn.br

**Data Availability Statement:** Model software is available at: https://github.com/Sly143/NatalCovid/.

## Abstract

The severe acute respiratory syndrome coronavirus 2 (SARS-CoV-2) pandemic hit almost all cities in Brazil in early 2020 and lasted for several months. Despite the effort of local state and municipal governments, an inhomogeneous nationwide response resulted in a death toll amongst the highest recorded globally. To evaluate the impact of the nonpharmaceutical governmental interventions applied by different cities—such as the closure of schools and businesses in general—in the evolution and epidemic spread of SARS-CoV-2, we constructed a full-sized agent-based epidemiological model adjusted to the singularities of particular cities. The model incorporates detailed demographic information, mobility networks segregated by economic segments, and restricting bills enacted during the pandemic period. As a case study, we analyzed the early response of the City of Natal—a midsized state capital—to the pandemic. Although our results indicate that the government response could be improved, the restrictive mobility acts saved many lives. The simulations show that a detailed analysis of alternative scenarios can inform policymakers about the most relevant measures for similar pandemic surges and help develop future response protocols.

**Funding:** PHL, LW, CRC, RCM, LMMV, RKRS, PSS and PAV received funding from Heriott-Watt University (832228-Singularity/COVID-19 Round 2019-20 GCRF-SFC). CRC received funding from The Royal Society and Newton Fund (NAF-R2-202209). The funders had no role in study design, data collection and analysis, decision to publish, or preparation of the manuscript.

**Competing interests:** The authors have declared that no competing interests exist.

## Introduction

The severe acute respiratory syndrome coronavirus 2 (SARS-CoV-2) pandemic started in Wuhan, China, in December 2019 [1–3] and quickly spread outside its borders, being acknowledged as a worldwide event by the World Health Organization in March 2020 [4]. The associated Covid-19 disease has a case fatality ratio of around 1.5% and is substantially more deadly for the elderly [5–7], imposing significant pressure on public health systems. By the end of 2021, the pandemic had claimed the lives of about 5.4 million people worldwide according to the WHO or up to 18 million casualties when considering excess death estimates [8]. Effective vaccines were only widely deployed in 2021, with a delayed distribution in underdeveloped countries [9, 10]. The transmission of SARS-CoV-2 is mainly airborne, with a relatively high estimated ratio of transmissions originating from asymptomatic or presymptomatic infected people [11–14]. Since testing availability differs significantly across countries, and because tests are only effective for a limited window of the infectious cycle, it is difficult to identify the infectious vectors. It is, thus, considerably challenging for public health operators to establish effective mitigation policies. Different countries applied a myriad of nonpharmaceutical interventions to reduce the impact of the pandemic, such as the closure of businesses and public services, the obligation to wear face masks, and limitations in mobility (lockdown). However, the inability to assess their effectiveness and the critical economic and social impact of such measures imposes substantial political costs to policymakers, who, in turn, often opt to drop them based on nonscientific arguments.

Establishing an optimal response strategy to the pandemic is especially problematic in Brazil —the largest and most populated country in Latin America, with over 211 million inhabitants [15]. The virus was introduced to Brazil from Europe between February and March 2020 [16]. After an early phase with locally constrained spread, the pandemic affected all regions of Brazil, with the first peak of deaths in June 2020 followed by a slow decay that reverted in November 2020. By the end of 2021, over 600 thousand deaths were reported in Brazil, with a peak 7-day average daily casualties rate above 3,000 deaths per day in April 2021 [17]. The pandemic in Brazil had two preeminent waves marked by high contamination levels and mortality, with more robust social mobilization in the first wave and higher impact rates in the second wave [8].

It is hard to evaluate the impact of the contention measures applied at a national level due to the heterogeneity of local responses. Brazil is a federal republic divided into 27 federation units (26 states and one federal district) and 5,568 cities. The Brazilian constitution assigns to the state governors and mayors the obligation to define sanitary measures in events such as a pandemic, with coordination from the Federal government through the Ministry of Health. However, political differences between the federal and local governments resulted in each state and city following an isolated agenda. Therefore, considering that the Federal Supreme Court recognised the right of each municipality to determine its own policies and given the extensive differences within single states—the State of São Paulo has a similar population size to Spain [18] —it is more beneficial to evaluate the impact of the contention measures at a municipal level.

Epidemiological models can help predict the impact of contention measures at a municipal level, but there are limitations. A key factor is that different cities have different mobility and hospital structures. In Brazil, most cities do not have a high-density mobility system, and there are separate private and public health systems. Also, most Brazilian urban centres have fairly isolated wealthy neighbourhoods contrasting with highly dense, poor districts with no sanitation. Importantly, current modelling strategies highlight that such characteristics significantly impact model performance. For instance, one of the first models focused on the Chinese city of Wuhan, a city with more than 11 million people, large mass transportation systems, hot summers, and cold winters [19]. A later modelling study focused on US temperate regions, with the authors

 

stressing that epidemics in tropical areas can be much more complex [20]. Also, the landmark model from Imperial College [21], used as a reference worldwide, emphasises that their results relate to US and UK data, with possible extension to high-income countries. Therefore, epidemiological models inspired by high levels of metropolitan transportation and more socially homogeneous societies might not reflect the epidemic dynamics of small and medium cities in underdeveloped countries. Considering the exponential dynamics common to epidemiological models, minor amendments to parameters or model architecture may incur a significant error with a severe impact on the health system and economics. Moreover, for the reasons mentioned above, nationwide epidemiological models present limited information for policymakers.

Here we use computational epidemiological modelling to assess the impact of governmental nonpharmaceutical interventions in the City of Natal, Brazil. The City of Natal is the capital of the State of Rio Grande do Norte, in the northeastern region of Brazil. With a total of 890 thousand inhabitants estimated in 2020 [15], Natal is among the 20 most populated cities in Brazil, located entirely in an urban area, with a territorial extension of 167.401 km$^2$, and a demographic density of 5,325.8 inhabitants per km$^2$ (6th largest among Brazilian capitals). The municipality stands out as an important Brazilian tourist destination due to its beautiful beaches, lagoons, and dunes, receiving around 2 million visitors annually from other parts of Brazil and the world. In economic terms, the service sector stands out in the municipality, with the Gross Domestic Product (GDP) being the 16th in the ranking of the capitals of the 27 federation units of the country [18]. In terms of well-being, Natal was, in 2010, among the 100 Brazilian municipalities with the highest Human Development Index (HDI) (0.763) [22]. However, the municipality has important social vulnerabilities that seem to have been amplified with the sanitary emergency, as the unemployment rate reached 13.8% in the first three months of the pandemic [23] while the most rigid social distancing prevailed. We have chosen the City of Natal due to the availability of well-documented epidemiological and geographical data. It shares similarities with many other Brazilian cities, such as population size, urban organisation, and policies undertaken during the pandemic.

In contrast to the commonly used compartmental models [24–27], we built an agent-based model that allows the inclusion of detailed demographic information that is commonly available, enabling easy replication of the analysis for other cities. Furthermore, agent-based models are superior in capturing complex heterogeneous urban, and social interactions during an infection outbreak [28, 29]. The model represents social interaction probabilities as graphs that display similar properties to census data. Such a complex network approach allows segmenting the modelled interaction between age groups and social modalities, such as religion, work, and school, to assess the impact of individual measures. Our strategy establishes a baseline model that reproduces the observed case fatality curve. Next, we modified model parameters to emulate alternative scenarios where the administration took different actions and evaluated the attainable fatalities outcome. Through this strategy, we can quantify the impact of lives saved or lost if the government had applied other policies.

Our results indicate that the policies enforced by local government could be significantly improved but nevertheless prevented a much more catastrophic scenario. While early decisions to close schools and universities saved many lives, reopening commerce and religious gatherings came with a substantial cost of lives.

## Materials and methods

### Epidemiological and demographic data

Epidemiological data of the pandemic in the State of Rio Grande do Norte (RN) has been made available online by the State Secretary of Health (SESAP-RN). The daily bulletins report

the anonymous data as age and gender of all confirmed cases, tests, hospitalisations, and deaths. For this study, we considered only the first epidemiological wave, defined here as two weeks before the first reported case on March 12th, 2020, to November 4th, 2020, when the average 7-day death rate was below 1 per day for the first time. Data of hospitalisation and ICU beds occupancy are also available from a different source of SESAP-RN (Regula-RN system). No information was available about the place of residence of the ICU occupants.

The local agency for urbanism and environment provided the total number of residents and age distribution for the City of Natal on the 2017 urban plan [30]. We collected demographic data from official governmental sources, which we used to describe the main social activities affected by the social distancing decrees during the pandemic, such as home, work, transport, religion, and primary and higher education. To describe the interactions within the household, we used the total number of family members calculated from the Continuous National Household Sample Survey—Continuous PNAD, a survey of national coverage and representative for Brazilian capitals and states [23]. Considering the workplace as an important space for disseminating the virus and directly affected by the decrees in social distancing, we used data from the Annual List of Social Information—RAIS [31]. In Brazil, employers provide the administrative records of workers in the formal sector to the Ministry of Economy. In this work, we considered five economic sectors: agricultural, industrial, construction, commerce, and service [31]. To describe the circulation activities, we took into account information from the last demographic census to obtain the average time the municipality residents spent for work [32].

Furthermore, information related to the carrying capacity of public transport in the City of Natal, provided by the Municipal Department of Urban Mobility [30], was used. The 2010 Demographic Census was also used to obtain the proportion of people who declared themselves Catholics and Evangelicals. These two religious categories were considered because they are, respectively, the most predominant in the Brazilian population [32]. Concerning educational establishments, Brazil has two sources of official census information: (1) the Brazilian School Census, from which we obtained the total number of students enrolled from kindergarten to professional education in public and private education in the City of Natal [33]; and (2) the Higher Education Census, from which we obtained information on the total number of students at the Federal University of Rio Grande do Norte on the Natal campus [34].

All demographic data is summarised on Table 1.

## Ethics statement

According to Brazilian Law, the study does not require an Ethical Committee evaluation as all data is anonymised and in the public domain.

## Epidemic agent-based model

We developed an agent model that extends the classic SIR model [35] with a more specific list of health states and with an agent-level individual interaction mechanism that implements social layers as complex networks [36]. This algorithm has two simulation parts: (1) the infection model implements a state machine that tracks the health of each agent across the simulation with respect to the expected disease course; and (2) a complex interaction network that simulates the agent interactions within multiple social layers (e.g. contact at work, transport, schools, etc.). In addition to that, we simulate the City of Natal population (a total of 873,383 agents) over 253 days (February 26th to November 4th of 2020), the first wave of the pandemic. This section summarises the simulation implementation. The simulation code, data, and user documentation can be found at Github.

**Table 1. Demographic data used for the model and their source.**

| Religion Data [32] | | | Home Data [23] | | |
|---|---|---|---|---|---|
| Age | Catholic | Evangelic | Household size | Occurance | Relative Frequency |
| 5 e 6 years | 13730 | 5058 | 2 | 72945 | 31.25% |
| 7 a 9 years | 22572 | 8720 | 3 | 73586 | 31.52% |
| 10 a 14 years | 41256 | 17637 | 4 | 53080 | 22.74% |
| 15 a 19 years | 46389 | 15726 | 5 | 22838 | 9.78% |
| 20 a 24 years | 54405 | 16279 | 6 | 7688 | 3.29% |
| 25 a 29 years | 51899 | 15599 | 7 | 2593 | 1.11% |
| 30 a 39 years | 83347 | 26798 | 8 | 964 | 0.41% |
| 40 a 49 years | 76641 | 21769 | 9 | 407 | 0.17% |
| 50 a 59 years | 54299 | 13421 | 10 | 151 | 0.06% |
| 60 a 69 years | 33869 | 8228 | 11 | 58 | 0.02% |
| | | | 12 | 32 | 0.01% |
| | | | 13 | 51 | 0.02% |

| UFRN Data [34] | | | | Transport Data [32] | |
|---|---|---|---|---|---|
| Undergrad Students | Postgrad Students | Teachers | Admin Staff | Average Time (min) | Number of People |
| 24680 | 6482 | 2344 | 2986 | 2.5 | 23080 |
| | | | | 12 | 135164 |
| | | | | 45 | 85013 |
| | | | | 90 | 21757 |
| | | | | 120 | 1613 |

**Work Data—Number of Workers by Sector [31]**

| Age | Agriculture | Industry | Construction | Commerce | Services |
|---|---|---|---|---|---|
| 15 a 19 years) | 209 | 1819 | 988 | 4810 | 7731 |
| 20 a 24 years | 253 | 6121 | 2683 | 3112 | 23268 |
| 25 a 29 years | 301 | 6300 | 3410 | 12602 | 29399 |
| 30 a 34 years | 285 | 5442 | 3258 | 11626 | 28344 |
| 35 a 39 years | 218 | 4922 | 3158 | 9204 | 24617 |
| 40 a 49 years | 345 | 4352 | 2968 | 7368 | 21671 |
| 50 a 54 years | 227 | 3180 | 2040 | 4684 | 16832 |
| 55 a 59 years | 313 | 2205 | 1408 | 2892 | 10533 |
| 60 a 64 years | 119 | 1131 | 781 | 2088 | 6368 |
| 65 a 69 years | 130 | 470 | 428 | 835 | 2266 |

**School Data—Students [33]**

| Public Kindergarden | Private Kindergarden | Public Elementary | Private Elementary | Public Professional | Private Professional |
|---|---|---|---|---|---|
| Age/Number | Age/Number | Age/Number | Age/Number | Age/Number | Age/Number |
| 0/0 | 0/1 | 10/5574 | 10/4597 | 20/695 | 20/742 |
| 1/108 | 1/101 | 11/5914 | 11/4622 | 21/433 | 21/599 |
| 2/1020 | 2/715 | 12/5851 | 12/4072 | 22/307 | 22/508 |
| 3/2712 | 3/1872 | 13/6541 | 13/4157 | 23/216 | 23/443 |
| 4/3835 | 4/3303 | 14/7008 | 14/3953 | 24/207 | 24/427 |
| 5/4701 | 5/4276 | 15/7194 | 15/3463 | 25/153 | 25/368 |
| 6/3570 | 6/2984 | 16/6909 | 16/2979 | 26/153 | 26/350 |
| 7/0 | 7/45 | 17/6798 | 17/2673 | 27/135 | 27/298 |
| 8/5433 | 8/4596 | 18/5421 | 18/1173 | 28/139 | 28/237 |
| 9/5171 | 9/4648 | 19/958 | 19/590 | 29/106 | 29/215 |

**Agent dynamics.** In order to better emulate the SARS-CoV-2 infection cycle, we extended the classic SIR model with additional states:

- **S**usceptible—The agent is vulnerable to SARS-CoV-2 and may be infected at any time. Initially, all agents are in this state.

- **I**ncubated—The agent has been exposed to the disease, but is not yet exhibiting symptoms.

- **A**symptomatic—The agent has the disease past the incubation period but does not show any symptoms.

- **I**mmune—The agent was infected and recovered; in our model, we assume once an agent has been infected, they develop immunity and cannot be re-infected. The agent may also have a natural immunity to the disease.

- **S**ymptomatic Mild—The agent was infected and has mild symptoms such as cough and sore throat.

- **S**ymptomatic Moderate—The agent was infected and has moderate symptoms. In addition to cough and sore throat (but not exclusively), the agent has headache and fever. They do not need hospitalisation at this stage.

- **H**ospital—The agent was infected, the symptoms continued to develop and they must be hospitalised.

- **I**CU—The agent is already hospitalised in critical condition and needs ICU and ventilation support.

- **D**ead—The agent developed severe symptoms and did not survive.

Initially, all agents are in a *Susceptible* health state. Agents get infected by close contact with another infectious agent by following the interaction rules of the complex networks (see below). Once infected, the health state of the agent turns into *Incubated*, starting a sequence of transitions through the different health states which will ultimately end as an *Immune* individual or a new *Death* (Fig 1A). For each infected agent, the model randomly assigns the most serious health state the agent will reach—and therefore also whether the agent will eventually die or not—following the probabilities that are presented in Table 2. The outcome likelihood is modulated by the age of the agent as reported by the State Secretary of Public Health (SESAP-RN) and were computed as the proportion of the number of confirmed cases by health state (*Symptomatic Mild, Symptomatic Moderate, Hospital, ICU* and *Dead*) for each age range. The probabilities for *Asymptomatic* cases were based on the proportion proposed by [37], resulting in a value of 18.8% that was applied to all confirmed cases.

The time an agent remains in the *Incubated* state is stochastic and follows a log-normal distribution of the agents' average disease incubation time [38]. The length of all other transitions is deterministic as they do not affect transmission dynamics but are defined based on the individual incubation time. As a result, a different proportion of health states develop in the population following the initial day of infection (Fig 1). On average, the agents spent four days in the *Incubated* state before evolving to *Asymptomatic*, where they remain for three days. Once at *Asymptomatic* health state, the agent may be able to become *Immune* or develop symptoms and reach the *Symptomatic Mild* health state, wherein it has to pass four days before becoming *Immune* [39]. However, already on the first day at this health state, the symptoms could develop and the health state changes to *Symptomatic Moderate*, in which the agent has to spend four days [40] before becoming *Immune*. If the symptoms become serious, the agent pursues *Hospital* health assistance. After one day at the hospital, the agent may be in critical

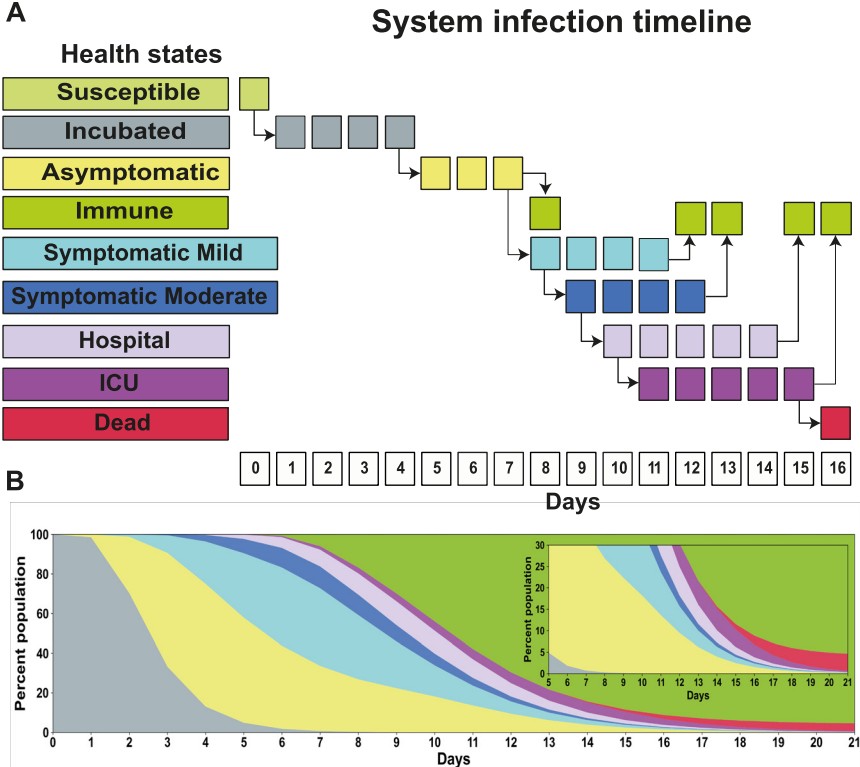

**Fig 1. Timeline of agent's health state progression through a SARS-CoV-2 infection.** (A) Possible daily transitions of one agent starting from *Susceptible* up to *Dead* or *Immune*. Sequences of the same state represent average time without change. (B) The evolution of the state of 10,000 agents, with age and sex distribution suited to the City of Natal, turned into *Incubated* at day zero. The simulation assumes illimited ICU bed availability.

condition and requires ICU support or ventilation; these require five days of intensive care. The time duration in hospitalisation and ICU was based on the State Secretary of Public Health (SESAP-RN). Importantly, the number of hospital or ICU admissions in the model is unlimited, i.e., all agents can be admitted at once if necessary.

Each agent is contagious in a specific infectious period that starts four days after the contamination (on average already at the *Incubated* health state) and finishes two days before the final incubation time in the *Asymptomatic* health state. If the agent progresses to *Symptomatic Mild*, it remains infectious up to two days before the end of this health state. These three health states are the only ones in which the agent can transmit the disease, which is an optimistic

**Table 2. Health state probability disease by age range.** Each health state is related to an age range, showing the outcome state that an agent reaches in case of infection.

| | | Age range | | | | | | | | |
|---|---|---|---|---|---|---|---|---|---|---|
| | | **0-9** | **10-19** | **20-29** | **30-39** | **40-49** | **50-59** | **60-69** | **70-79** | **80+** |
| Final stage of Health | *Asymptomatic* | 0.1579 | 0.1587 | 0.158 | 0.1583 | 0.1582 | 0.1584 | 0.1583 | 0.1583 | 0.1583 |
| | *Symptomatic Mild* | 0.4269 | 0.5904 | 0.6226 | 0.6088 | 0.5647 | 0.4421 | 0.3442 | 0.1883 | 0.106 |
| | *Symptomatic Moderate* | 0.0819 | 0.1513 | 0.1682 | 0.1348 | 0.1041 | 0.0854 | 0.0536 | 0.0215 | 0.0102 |
| | *hospital* | 0.3333 | 0.0996 | 0.0512 | 0.981 | 0.173 | 0.3142 | 0.4439 | 0.6319 | 0.7255 |
| | *ICU* | 0.0643 | 0.0185 | 0.0096 | 0.18 | 0.0412 | 0.0961 | 0.1603 | 0.2847 | 0.3355 |
| | *Dead* | 0.0234 | 0.111 | 0.0084 | 0.0135 | 0.0292 | 0.0742 | 0.1147 | 0.2221 | 0.2752 |

**Table 3. Social interaction layers information.** The first column (Layer) shows the layers (total of 7) implemented on the system, and sub-layers (schools, work and religion), that further describe some of the activities (second column). The third column (Agents age) represents the age range distribution for each layer/sub-layer. The fourth column represents the probability of agents belonging to the layer. In some sub-layers (schools and work), the sum of this probability is 100%, which means that all agents within an age range must belong to only one of these sub-layers. The fifth column (Time per week) is the average time the agents spend interacting with other agents on this layer/sub-layer. The sixth column (Nearest) shows the agent's average number of contacts per layer/sub-layer. The last column (Group size) represents the average group size and the agent's distribution; per layer/sub-layer.

| Layers | Sub-layers | Agents age | Belongingness probability [%] | Time per week ($T_w$) | Nearest ($K_i$) | Group size ($G_s$) |
|---|---|---|---|---|---|---|
| Schools | Kindergarten public | 0-9 years | 54.53 | 20 hours | 8 | [10-55] Norm |
| | Kindergarten private | 0-9 years | 45.47 | 20 hours | 2 | [3-24] Norm |
| | Elementary public | 10-19 years | 62.36 | 20 hours | 7 | [8-32] Norm |
| | Elementary private | 10-19 years | 37.64 | 20 hours | 4 | [11-19] Norm |
| | Professional public | 20-29 years | 56.43 | 25 hours | 5 | [6-64] Log-norm |
| | Professional private | 20-29 years | 43.57 | 25 hours | 4 | [6-39] Norm |
| Work | Agriculture | 20-69 years | 0.87 | 40 hours | 5 | [6-41] Norm |
| | Industry | 20-69 years | 11.96 | 40 hours | 5 | [6-37] Norm |
| | Construction | 20-69 years | 7.12 | 40 hours | 5 | [6-38] Norm |
| | Commerce | 20-69 years | 22.91 | 40 hours | 4 | [5-35] Norm |
| | Services | 20-69 years | 57.13 | 40 hours | 5 | [6-37] Norm |
| Home | Home | Everyone | Everyone | 21 hours | Everyone | [2-13] |
| Transport | Transport | 10–80+ years | 31 | 1h44min | 2 | [3-70] Uniform |
| Region | Catholic | 0-69 years | 54.77 | 2 hours | 8 | [17-100] Norm |
| | Evangelic | 0-69 years | 17.08 | 2 hours | 9 | [18-100] Norm |
| UFRN | UFRN | 10-69 years | 4.1 | 40 hours | 5 | [12-180] Norm |
| Random | Random | Everyone | 5 per person | 1 hour | 1 | 1-to-1 |

assumption because it considers that 1) once the agent reaches the *Symptomatic Moderate* it will be isolated, and 2) that the agents that progress to *Hospital* will not infect the health staff.

**Complex networks of social interactions.** At each cycle of the simulation (day), the system computes the possible social interactions between agents of the population to emulate the contagious aspect of the pandemic. As a result of this interaction, one agent in the infectious period can infect another susceptible agent and change its health state into incubated. The probabilities of interaction are determined by multiple complex networks of social interactions (layers and sub-layers) that include an agglomerate of agents. Initially, when the agents are created, they are distributed to the layers/sub-layers according to the age range (Table 3, third column) and the probability of belongingness (Table 3, fourth column). The time per week during which social interactions happen in each layer/sub-layer is displayed in the fifth column. Once allocated, they are separated into sub-groups to represent the many activity clusters. The average direct contact among agents and the group size is presented in columns six and seven, respectively. Each network has its specificities and is further detailed in Table 3. The probability of agents interacting among themselves (referred throughout the text as $P_{interaction}$) within layers/sub-layers is calculated over time as follows. Consider two agents $A_1$ and $A_2$ belonging to the same sub-layer, its interaction value is calculated by:

$$P_{interaction}(A_1, A_2) = \left( \frac{T_w}{168} * \frac{K_i}{G_s} \right) * P_{contamination} \tag{1}$$

Where $T_w$ is the time per week spent by the agents in the sub-layer, $K_i$ is the average of direct contacts with other agents, 168 is the total hours in one week (24 hours * 7 days = 168 hours) and $G_s$ is the group size inside the sub-layer. The $P_{contamination}$ value represents the probability of infection upon contact, i.e., the probability of virus spread. Initially, this value is 1.7 and was

## Complex network layers and connections

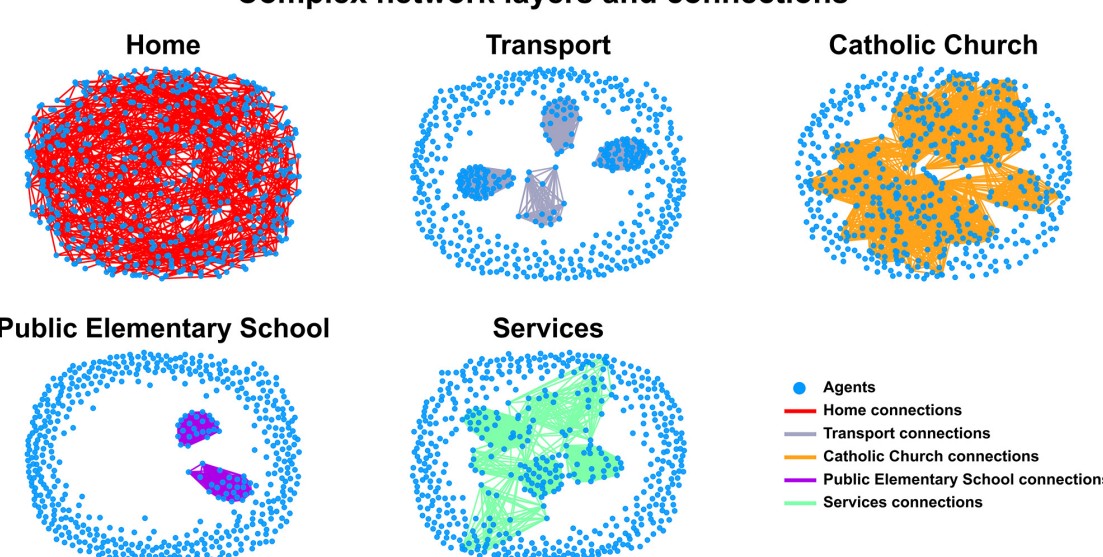

**Fig 2. High diversity of contact networks.** Layers and sub-layers are complex networks composed of agents (blue dots) and social interactions (lines). Representative layers (Home and Transportation) and sub-layers (Catholic churches, Public Elementary Schools and Services) display different characteristics as high connectivity, coverage and a small world topology. All the information about connections is available in Table 3.

obtained from (SESAP-RN) as the average of $R_0$ value for the first wave. Each layer/sub-layer has its own $P_{contamination}$ and this value is modified over the days as a consequence of the decrees (more details are presented in the Decrees sub-section).

These complex networks of social interactions were built based on demographic data detailed in Table 3. The structural differences among the different layers and sub-layers become evident by plotting the agents and their social interactions (Fig 2). To better represent the social interaction of the City of Natal population, we divided some activities into sub-layers, such as Work, School and Religion (Table 3, second column). The first six layers (Agriculture, Industry, Construction, Commerce and Services) belong to the "Work layers" group. The following six layers (Kindergarten, Elementary and Professional Education; public and private) correspond to "Schools layers". The last five layers (Home, Random, Transport, UFRN, Catholic and Evangelic) belong to the "Other layers" group.

The School layer was split into six sub-layers to represent the educational system in Natal. This layer is composed of: public and private kindergarten, elementary, and professional education. Each agent may only be a part of one sub-layer according to their age range (Table 3, the third column). The fourth column shows the probability of an agent belonging to each sub-layer. These probabilities add to 100% in each educational age range for public and private. All groups follow a normal distribution, except the Professional public education layer, where the individuals in this layer have a log-normal distribution. The data used in this layer comes from 2019 Brazilian School Census [33].

To represent the Work layer, we considered formal employment in five economic sectors: agricultural, industrial, construction, commerce, and service. The work activities were implemented as a sub-layer where participation is exclusive, i.e., an agent can belong to only one sub-layer (Table 3, the third column). Agents in these sub-layers are aged between 19 and 69

years old and are assigned using a normal distribution. The data used to create this layer are from the 2018 edition of the RAIS [31].

For the creation of the Home layer, the distribution of the Brazilian family size was calculated based on data from PNAD for the fourth quarter of 2019 and the first quarter of 2020 [23], followed by the estimation of the household size probability (Table 1). The distribution was then used to assign each agent to a household of 2 to 13 people.

To model the transport layer, we considered only public transportation, since it is one of the most populated environments, in which each vehicle has a passenger capacity of 70 people [30]. To create this layer, we used data from the 2010 Brazilian Demographic Census [32], which presents data on the time spent on the journey from home to work or school. Here we used the same methodology as [41, 42].

Churches are another high-risk environment for the spread of viruses. Here we used two layers for Natal's most common religions: Catholic and Evangelical. We consider a maximum capacity of 100 people between 0 and 69 years old with a single 2 hours gathering per week. We assigned both layers using a normal distribution. The data for this layer was obtained from the 2010 Brazilian Demographic Census [32].

A separate layer regarding higher education refers to the Federal University of Rio Grande do Norte (UFRN) because it has the largest number of students and workers in the City of Natal (groups of individuals from 12-180). This layer has agents from almost every age range (10-69 years old) and approximately 4.1% of Natal's population has some connection with UFRN. We used a normal distribution to assign agents to this layer, and the data used comes from the Higher Education Census [34].

The Random layer was implemented to represent any other direct contact between agents, such as drug stores, markets, public parks, etc; or indirect contact, such as through objects, surfaces, etc. In this layer, agents are randomly connected regardless of the age range.

**Decrees.** The government's decrees were the primary tool of local government action to modulate the pandemic course. Decrees included the closure of specific economic and social parts of the society and the obligatory use of masks. Within the model, we implemented the decrees as a change in the probability of contamination ($P_{contamination}$) and the level of agent interactions ($P_{interaction}$) that are specific to layers and sub-layers over a particular period. Although all government decrees have their epidemiologic importance, we implemented only the most impactful model adjustments (Table 4). For example, the decree on day March 25th named "Alecrim closure" was an important factor because this neighbourhood is the central

**Table 4. Decrees affect simulation dynamics with variable impact on the different layers and sub-layers over the first wave pandemic.** The first column represents the decree's name. The second column is the implementation day. The third column displays the layers and sub-layers affected by the decree. The baseline simulation scenario defines the $P_{contamination}$ value relative to each decree.

| Decree | Day | Affected layers |
|---|---|---|
| Close schools | March 17th | UFRN, Professional (public and private), Elementary (Public and Private), and Kindergarten (Public and Private) |
| Partial quarantine | March 20th | Agriculture, Industry, Construction, Services, Commerce, Catholic, Evangelic, Home, Transport, and Random |
| Alecrim closure | March 25th | Agriculture, Industry, Construction, Services, Commerce, Transport, and Random |
| Reopening commerce and transport | April 9th | Agriculture, Industry, Construction, Services, Commerce, Transport, and Random |
| Mandatory face masks | May 5th | Agriculture, Industry, Construction, Services, Commerce, Catholic, Evangelic, Home, Transport, and Random |

commercial area of Natal city. We did not set the $P_{contamination}$ value to zero because some establishments continued to work.

**Estimation of external Covid-19 cases and ICU beds.** The pandemic starts with initial infections of local individuals by external agents. To estimate the number of daily new infections during the first wave pandemic in Natal city due to external interactions, we used daily information about flights and buses, estimated highway traffic and daily reports of confirmed cases.

Natal has only one bus station with an estimated daily number of passengers of around 2500 [43]. The pandemic and the March 20, 2020 decree impacted the bus flow with a decrease by 50% [44]. Until the end of December of 2020, the number of passengers slowly increased to around 90% of the regular flow [45]. Thus, we estimated the total number of bus station passengers during the first wave was 367,875.

The number of flights in Natal airport followed the Brazilian pattern. After the restriction measures were established in March 2020, an agreement was reached between the aviation companies and ANAC (National Civil Aviation Agency) [46], with a minimum number of flights between the capitals being defined due to the economic infeasibility of maintaining flights with a reduced number of passengers. With the sanitary and financial measures adopted by the government, the number of flights has been slowly returning to normal, reaching 70% of the usual number of flights in December 2020 [47]. Therefore, the number of estimated passengers during the pandemic's first wave was around 452,233.

Natal is the capital of the state and a metropolitan city. Although we do not have access to the highway flow during the first wave, this number was estimated at 50% of the total cases previously calculated. According to (SESAP-RN), the sum of Covid-19 confirmed cases in the first wave was 26,371 (Fig 3A, red line). We used the data to mould the estimated daily external infections (Fig 3A, blue line). First, the daily percentage of infection was obtained by dividing the daily confirmed infection cases by the number of the Natal city population. After that, this percentage was applied to the daily sum of passengers from the bus station and flights, as shown in Eq 2.

$$EI_{[day]} = CC_{[day]} * \frac{TP_{[day]} + AP_{[day]}}{N} \qquad (2)$$

Where $EI_{[day]}$ is the daily external infections, $CC_{[day]}$ is the confirmed cases by day, $N$ is the total population of Natal, $TP_{[day]}$ is the daily terrestrial passengers and $AP_{[day]}$ is the daily air passengers. Thus, the Eq 3 calculated the total external new cases into the model (*TEI*) by adding the estimated number of external infected through highway flow (HF) and the sum of $EI_{[day]}$ calculated in Eq 2. In summary, among all agents of the Natal population (total of 873,383), a total of 3,957 (the amount of new external cases) are randomly chosen. These picked agents have their health state changed to *Incubated*. These cases were modelled as a Gaussian distribution, following the first real infected confirmed case date on March 12th 2020 (Fig 3A, blue line). We opted to not use the real number of confirmed cases due to the large absence of tests capable of detecting the Covid-19 pathology in Natal city.

$$TEI = EI_{[day]} + HF \qquad (3)$$

To estimate the ICU bed availability and occupancy for the City of Natal during the first wave of the pandemic, we considered the population size ratio between the metropolitan region and the Rio Grande do Norte state (RN). This metropolitan region is composed of 14 cities, such as Arês, Ceará-Mirim, Extremoz, Goianinha, Ielmo Marinho, Macaíba,

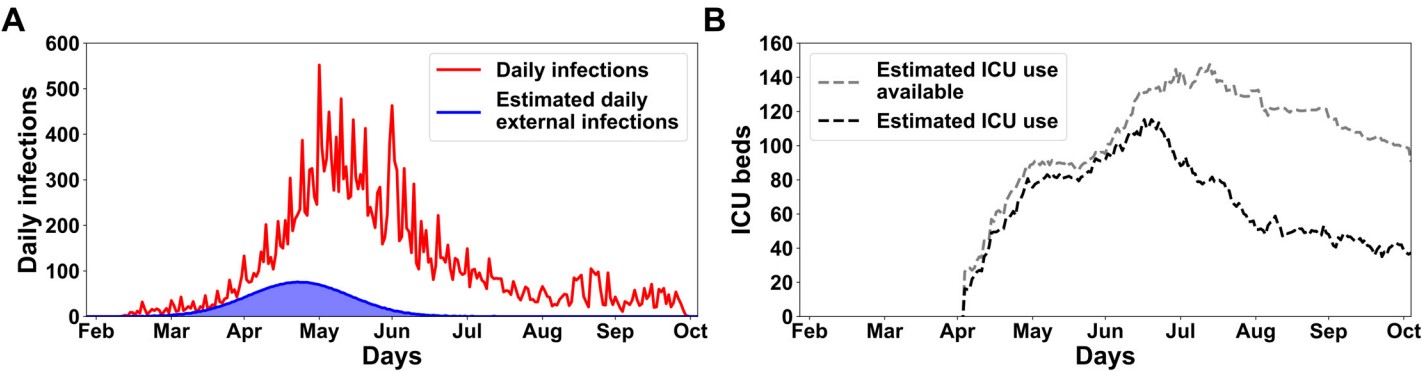

**Fig 3. Epidemiological data on the first wave of the Covid-19 pandemic in the City of Natal, Brazil.** (A) The daily number of confirmed cases with a total of 26,371 cases and a peak of 552 new cases in one day (red line), and the estimated daily number of external cases with a total of 3,957 cases with a peak of 76 new cases (blue line and area). (B) Estimated ICU beds available (silver dashed line) and utilized (black dashed line) during the first wave. Numbers for Natal are estimated as 46.68% of the metropolitan region data.

Maxaranguape, Monte Alegre, Natal, Nísia Floresta, Parnamirim, São Gonçalo do Amarante, São José de Mipibu and Vera Cruz. The entire RN state has a population of approximately 3,419,010 people, and the metropolitan region has around 1596103 inhabitants according to [30]; thus, the metropolitan population proportion is 46.68%. This proportion was applied to the actual data acquired from the website (RegulaRN system), which monitors the number of ICU beds available and occupied over the days; for the RN state. The real data were decreased proportionally (46.68%) with respect to the metropolitan region population (Fig 3B).

## Estimation of baseline scenario

Estimating the baseline scenario consists of defining the values for the open variables that produce a simulation that best reproduces the actual report of deaths. The open variables are (1) the initial $P_{contamination}$ value of each layer/sub-layer and (2) the change in $P_{contamination}$ value implemented by each government decrees. Each decree increases or decreases the $P_{contamination}$ value of different layers/sub-layers, on the day of the decree implementation, thus changing the course of the pandemic simulation. To evaluate the quality of a baseline candidate parameter set, we used the R-squared [48] difference between the simulated and the reported curve of accumulated deaths and the absolute difference between the mean cumulative deaths of the model outcome and the real data.

We implemented a multiphase search algorithm to find the most appropriate parameter set for the baseline scenario. First, we computed the R-squared value and the absolute difference to the accumulated number of deaths for a set of 10,000 randomly assigned parameter sets for a single run. The 100 best candidates (sorted by R-squared above 0.95 and accumulated death value with <10% difference to actual reports, 11 deaths) continued for a 30-run average. Next, we performed a local search for the best candidate by varying each parameter on nine other values in its vicinity (parametric steps of +/- 0.8, 0.5, 0.2, and 0.1). We selected the parameter set with the lowest difference in death counts from the actual data considering the sets with an R-squared value above 0.99. We then simulated the baseline model for 500 runs to set the reference for the other scenarios.

# Results

## Analysis of sensitivity of the model to individual transmission layers

We built an agent-based model to simulate the pandemic evolution through the entire City of Natal with a total of 873,383 agents and multiple contact networks connecting agents. The contact networks emulate different layers of social interaction, including schools, work, religious gatherings, transport, commerce, and households. We modelled each contact network based on available urban data as a complex network with specific parameters (see Methods). Based on these networks, the simulation computes the contacts that drive the transmission of the disease. The model's only set of free parameters, $P_{contamination}$ (see Methods), determines each network's relative "temperature" with an impact on the total number of contacts in each epoch. I. e., higher $P_{contamination}$ values will lead to a higher number of connections following the dynamics of the specific complex network.

As we built all networks based on data gathered from public repositories and requests from local governments, we first produced a sensitivity analysis to assess the baseline impact of each network in the primary simulation outcome: the accumulated number of deaths (Fig 4). For this analysis, we simulated the model with all networks set with homogeneous $P_{contamination}$ values (1.7), which led to a median of 29,023 deaths (28,736, 28,907, 29,121, 29,298, as the percentiles 5%, 25%, 75% and 95%, for n = 500). Next, we globally reduced the $P_{contamination}$ values to a lower standard (1.5) to set a reduction reference. We observed a median paired reduction of 1,613 deaths (1,586, 1,597, 1,623, 1,634), considering the same random seeds. Next, we run the high-value simulation, reducing each layer's specific $P_{contamination}$ value to a lower value (1.5),

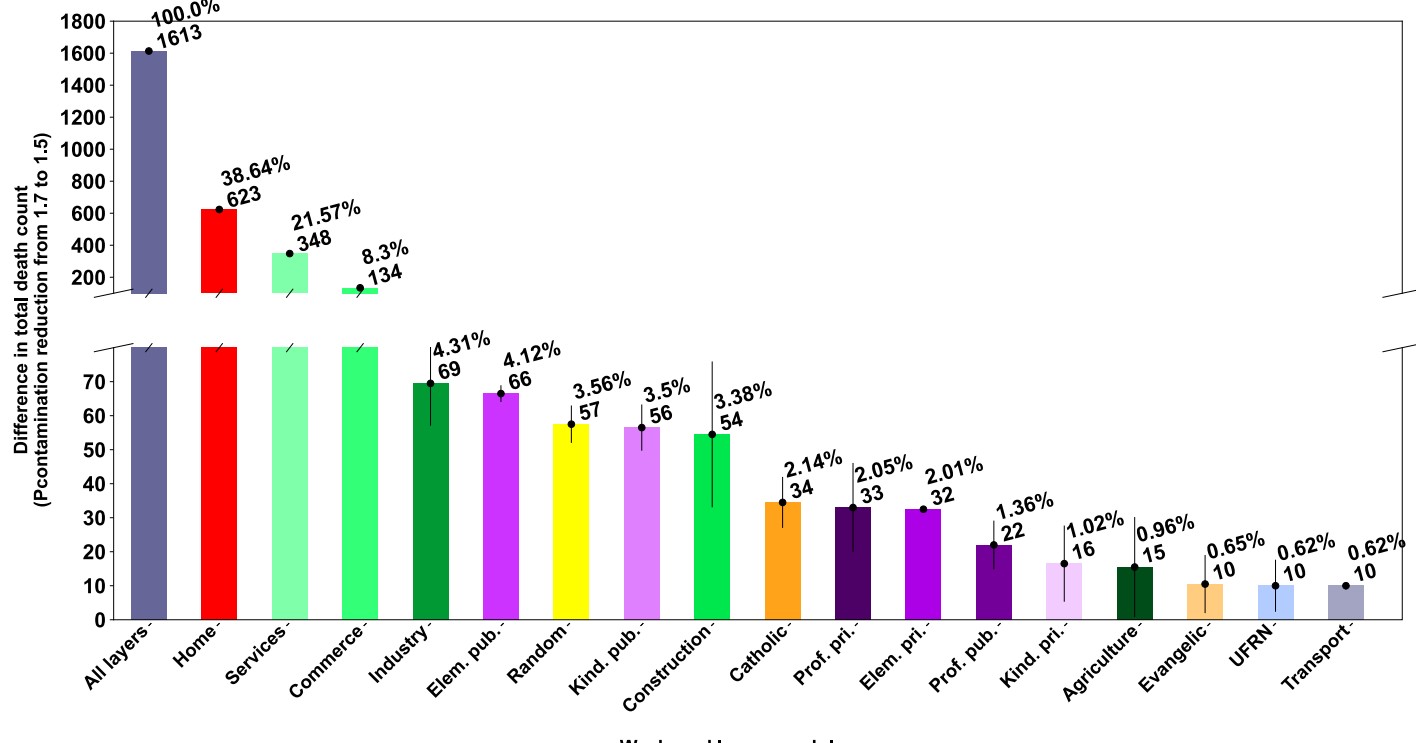

**Fig 4. Sensitivity analysis indicates an inhomogeneous impact of different transmission networks in the outbreak progression.** The difference in the total number of deaths with a reduction of $P_{contamination}$ from 1.7 to 1.5 in all layers, in each layer or sub-layer. Graphics represent median (bar), quartiles (line), absolute median difference to high-value simulation and relative difference to a reduction of $P_{contamination}$ value in all layers.

which led to a drop in the accumulated number of deaths that is a proxy for how sensitive the model is to each specific network. We found that the layers alone could respond from about 40% to less than 1% of the simulation outcome, indicating a very heterogeneous influence of different networks in the transmission dynamics.

## Search for a baseline model of the pandemic's first wave in the City of Natal

The first confirmed case of SARS-CoV-2 in the City of Natal was on March 12th 2020 [49]. The number of cases increased and peaked on June 1st 2020, with a 7-day average of 552 new cases per day. The first death was reported on March 31st 2020 [50]. Fatalities follow the growth of new cases to a peak on June 1st 2020, with a 7-day average of 16 deaths per day. The death rate reduced to a 7-day average of new deaths below one event per day in September 2020, returning to an increasing trend in early November 2020. From then on, the first day with no reported deaths by SARS-CoV-2 was October 2021. This analysis only considers the first wave (starting on February 26th, two weeks before the first confirmed case, and ending on November 4th). During this wave, the Natal city Secretary for Public Health reported 1072 deaths and 26,371 confirmed cases. We focused on the first wave as governmental action was consistently lower during the second wave that took effect in early 2021, as indicated by the lack of new specific regulatory actions. Moreover, it is impossible to draw a straight comparison between the waves as the second wave was caused by a different virus variant (P1/Gamma) [51], and the vaccine roll-out was concurrent with the second surge [52].

With the model in place, we searched for the set of free variable values ($P_{contamination}$, see Methods) that allowed the best replication of the reported accumulated deaths curve in the first pandemic wave in the City of Natal (Fig 5). We modelled the governmental decrees as a temporal change in the $P_{contamination}$ value, which was also subject to the search. Notably, as the population's adherence to the orders is unknown, we only zeroed the $P_{contamination}$ of the school layer, which is the only social layer where we could not find reported violations. Infections due to external agents were estimated from air, road traffic and confirmed infections curve (see Methods). The output baseline parameters set from the search algorithm was: Prior to decrees: 1.70; Partial quarantine = 0.86; Alecrim closure = 0.71; Reopening commerce and transport = 0.74; and Mandatory face masks = 0.63. After 500 simulations, the death cumulative difference value decreased to only 1, and the R-square was 0.9927.

The baseline model could adequately reproduce the daily and accumulated curve of deaths in the first wave (Fig 5A and 5B, respectively). In the initial months (Feb to early Jun), the simulation outcome followed the curve of actual death data. After June, the model results follow closely below the actual data until the start of October, finishing the simulation period with a median of five deaths above the actual data. In total, the observed accumulated number of confirmed deaths was 1072, while the median number of deaths in the simulation was 1,077 (quartiles = [982, 1, 040, 1, 109, 1, 152]) (Fig 5B). The simulations indicate 23,475 citizens were infected during the first wave, suggesting a sub-notification of about 10.98%. With the baseline model in place, we could inquire about the importance of the different layers for disease propagation by evaluating the amount of new infectious agents in each social network.

We also used the model to quantify the number of infections that originated in one specific layer and sub-layer (Fig 5C and 5D). Notably, there is a strong resemblance between the original layers of the infections and the impact measured in the sensitivity analysis, such as the high impact of home and services and the low incidence of transport-originated infections. Schools-based transmission is not observed because of the March 17th, 2020, decree that closed all schools (vertical black dashed line). It is also possible to estimate the use of ICU beds by the simulation (Fig 5E, purple line). In comparison to the estimated number of ICU beds

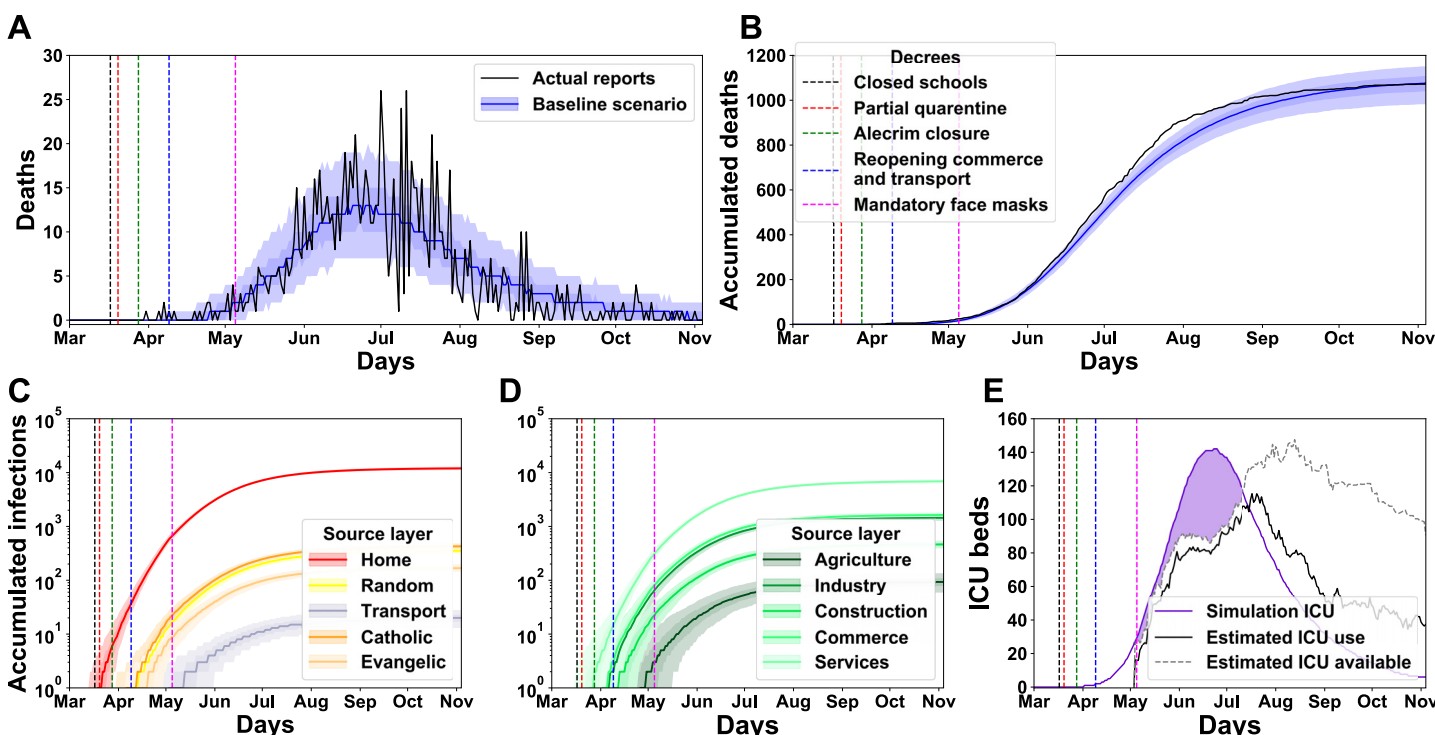

**Fig 5. Baseline simulation from the agent model provides a good fit for epidemiological data on the first wave of the SARS-CoV-2 epidemic in the City of Natal, Brazil.** (A) Daily and (B) accumulated deaths during the first wave of the SARS-CoV-2 outbreak in the City of Natal (from the end of February to the beginning of October) from simulation (blue) and actual reports (black). Vertical lines indicate the dates of publication of governmental decrees. (C and D) The accumulated number of infections originated in each layer or sub-layer. Simulation data from (A to D) were reported as median and quartiles (5%, 25%, 75%, 95%) from 500 runs. (E) Model-predicted ICU requirement (solid purple line, median) and excess (purple area) from estimated daily availability (silver dashed line) and the actual estimated occupation (black line) of ICU beds for the City of Natal.

available for Natal (silver dashed line) and the estimated number of ICU beds used (black line) during the first wave (actual data start on May 4th), one can observe some differences. The simulated ICU bed demand was above availability, with 7,119 non-available bed requests, whereas the real ICU bed demand had a peak of 142 people in line on June 19th. One possible interpretation for the observed discrepancy is that real ICU demand might have been higher than availability during this period, which might explain the brief difference between the simulated and real death toll in late July. Importantly, the number of ICU beds is unlimited in the simulation, and all critical cases, which needed ICU, were treated.

## Evaluation of the impact of individual decrees

To evaluate the impact of individual decrees, we explored different hypothetical scenarios, including (a) absent interventions, where some decrees have not been applied; (b) effective implementation of the decrees, where the affected layers/sub-layers were fully locked down; and (c) delayed interventions, where there was a delay in the implementation of the decrees. We considered three main classes of interventions focused on (1) school layers, (2) workplace layers and (3) religious layers. We computed the expected number of deaths for each scenario and compared it with the observed number of deaths in the baseline scenario. For the baseline, absent and effective scenarios, a total of 500 simulations were run. For the delayed intervention scenario, a total of 250 simulations were run for each delay step. We also considered a non-intervention scenario where all decrees were absent, with a total of 500 runs.

## Evaluation of school-related decrees

We investigated the impact of the education-related decrees in the containment of the pandemic (Fig 6). Two decrees enacted on March 17th, 2020, closed all the City of Natal educational institutions. We considered three hypothetical scenarios in which: no school was ever closed (Fig 6A); all schools closed after the ongoing term (Fig 6B); and there was a delay of a variable number of weeks in the date of decrees' publication (Fig 6C). We found a considerable increase in deaths in all scenarios compared to the baseline simulations. In the most impactful scenario without schools' closure, although the initial months (February, March and early April) had a death toll almost invariant to the baseline scenario, the following months produced a remarkable increase in mortality (Fig 6a.1 and 6a.2). Natal would have a median of 6342 cumulative deaths (6,023, 6,208, 6,467, 6,642, n = 500), an increase of 5,270 (490%) deaths compared with the actual reports (black line). Natal City reached the highest daily death value on July 1st, with 26 deaths reported. In such a disastrous scenario, the simulated daily deaths peak between August 13th and 18th, with a median of 51 daily deaths. An inspection of the origin of the infections in the simulation scenario without an educational decree revealed that the different educational institutions had a variable impact on the transmission of the disease (Fig 6a.3–6a.5). Moreover, the increase in infections would provide a demand for ICU units well above the available offer (Fig 6a.5). In a similar scenario, in which only the local federal university (UFRN)—which required a separate decree—would continue its activities regularly, we found a median increase of 44 deaths (40, 44, 54, 56) compared to the baseline scenario.

In the following scenario, we considered the closure of all educational institutions on the last day of the ongoing term, on June 29th, 2020 (Fig 6B, vertical black dashed line). This decree results in a considerable decrease in cumulative deaths compared with the previous scenario (Fig 6b.1 and 6b.2, indicating that even a late government intervention would still save lives. Still, the total number of deaths is above 50% of the observed in the baseline scenario. We found no considerable changes in the origin of the transmission (Fig 6b.3–6b.5). ICU demand would still be well above the offer (Fig 6b.5). We also investigated what would be the impact of shorter delays on the promulgation of the decree that closed all schools Fig 6c.1 and 6c.2). We found little impact in the number of casualties if the government delayed the closure of schools by four weeks with a 3% increase in deaths, but a fast increasing number of deaths if the decision-makes waited more than six weeks with a two-fold increase with a ten weeks delay (Fig 6c.2 insert).

## Evaluation of work-related decrees

Three decrees regulated work-related activities: one enacted on March 20th partially closed high-density businesses, another on March 25th closed all businesses and the last partially reopened the commerce on April 9th. Despite strong governmental efforts to oversee the implementation of the decrees, the local press reported many breaches. Thus, contrary to the school layers that the orders effectively shut down, the government could only weaken the work-related transmission networks. We modelled this aspect of partial adherence of the population to the decrees by setting the reduction o $P_{contamination}$ as an open variable for the work-related layers. For this analysis, we considered three scenarios (Fig 7): one without the second and most restrictive decree, which would let unaltered the routine of the most popular neighbourhood (Alecrim), another scenario in which the regulation effectively shuts all businesses, and the last scenario in which the government delayed the effective shut of business.

In the first scenario, without the March 25th decree that forced the closure of the Alecrim neighbourhood [53], the simulations indicated a moderate increase in the lethality of the first epidemic wave (Fig 7A). The accumulated number of deaths has a median of 1,240 (1,148,

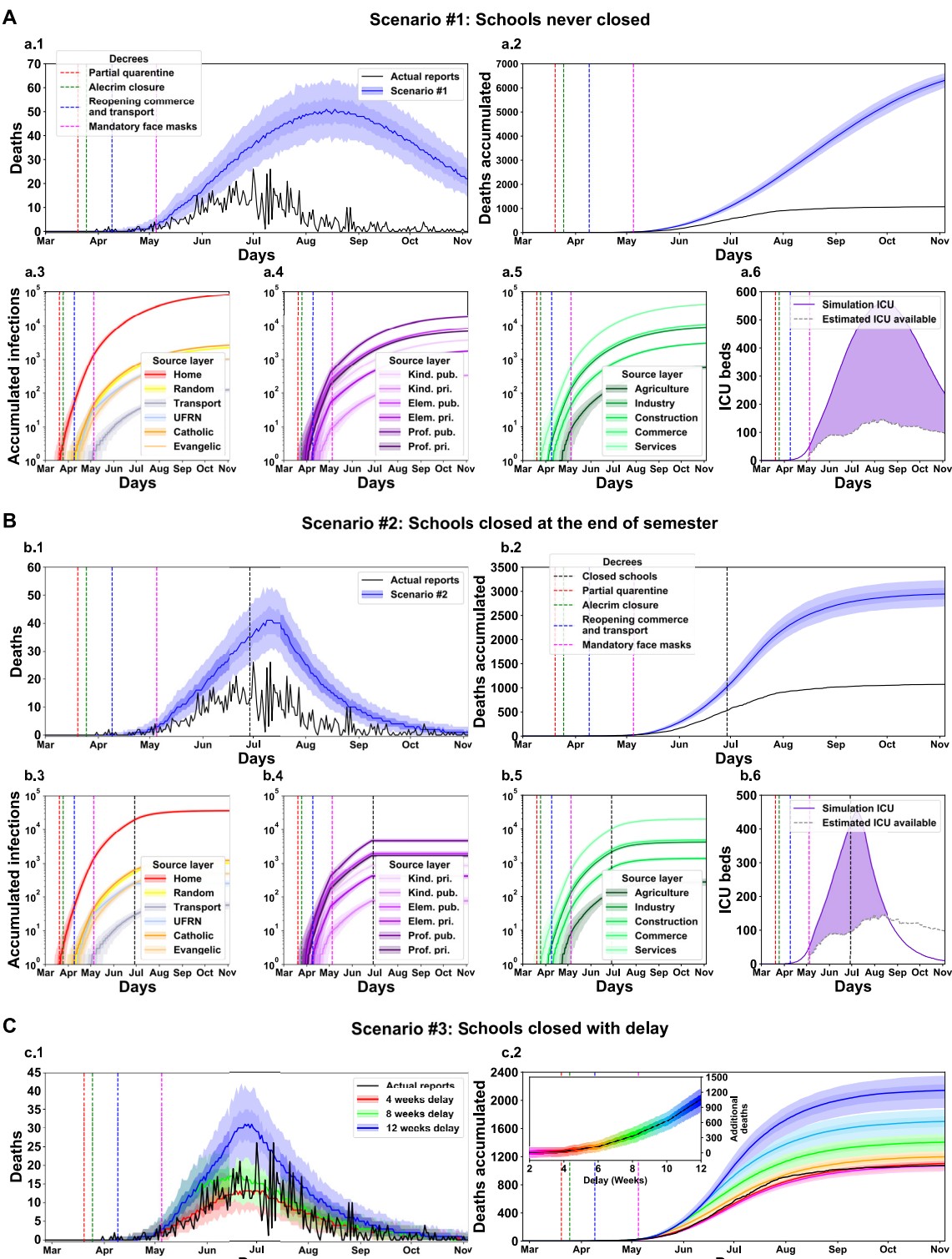

**Fig 6. Simulation of different scenarios reveals a significant impact of the school closure decree on March 17th, 2020.** Results for a baseline scenario altered for the condition that schools were never closed (#1, A), closed at the end of the semester (#2, B) and closed with a delay from the original decree's publication date (#3, C). The total number of additional deaths in each scenario is the difference between the simulation outcome and the reported number. For scenario #3, the delay in weeks is colour-coded, and the additional number of deaths is shown in an insert. See Fig 5 for panel description.

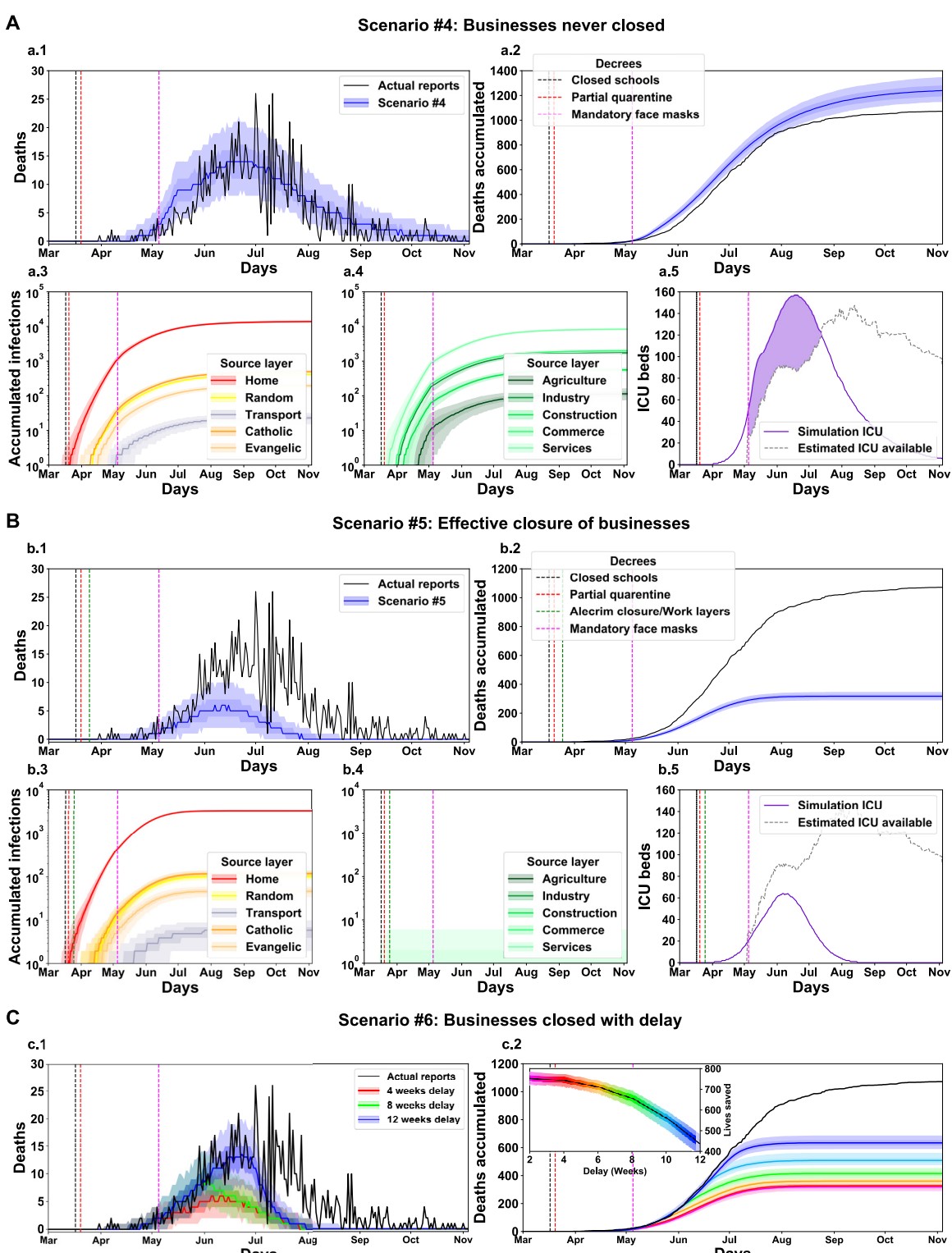

**Fig 7. Simulation of different scenarios reveals a moderate and sub-optimal impact of the business closure decree on March 25th, 2020.** Results for a baseline scenario altered for the condition that workplaces were never closed (#4, A), closed effectively (#5, B) and closed effectively with a delay from the original decree's publication date (#6, C). The total number of lives saved in each scenario is the difference between the reported number of deaths and the simulation outcome. For scenario #6, the delay in weeks is colour-coded, and the additional number of lives saved is shown in an insert. See Fig 5 for panel description.

1,193, 1,281, 1,349, Fig 7a.1 and 7a.2), which is an increase of around 15% in regards to the baseline scenario. The impact on the distribution of transmission origins (Fig 7a.3 and 7a.4) and the ICU demand (Fig 7a.5) is not exceptional. In the second scenario, we considered the situation where the decrees could effectively close all businesses (Fig 7B). We found a 3-fold reduction in the number of casualties (median 317, (288, 306, 329, 347), Fig 7b.1 and 7b.2). Even with the $P_{contamination}$ value set to zero in the work layers, the activity blockade could not avoid all the infections (Fig 7b.3 and 7b.4). The reason is that the enaction of the decree occurred only 16 days after the first confirmed case in Natal city. The substantial reduction of infections was an important factor in reducing the occupancy of ICU beds by the simulations. In this scenario, the ICUs peak was on June 5th (median total of 64) and did not reach the estimated Natal city ICU availability (Fig 7b.5). In the last scenario, we asked whether a delayed effective closure of work-related layers would still impact the number of lives saved (Fig 7C). We found that a delay of up to 3 months would still produce a positive value for lives saved. Therefore, it would be best to have a delayed but more strict implementation of the decrees than the early but not so stringent ones.

## Evaluation of worship-related decrees

Closing worship institutions was one of the most controversial governmental acts during the first wave, as religion is a relevant component of Brazilian culture. The local administration implemented it along with the decrees that closed other businesses. Importantly, just as in other businesses, there were reports of irregularly open churches. Therefore, we analysed the impact of closing churches and other worship venues in terms of lives saved following the same analysis methodology of the work-related decrees (Fig 8).

We analysed three scenarios: one in which the churches never closed (Fig 8A), another in which churches effectively closed (Fig 8B) and one in which churches closed effectively but with a delay (Fig 8C). The results are very similar to those found with work-related businesses but at a smaller scale. Our results indicate that the attempted closure of worship venues had a marginal effect on the number of deaths (Fig 8a.1 and 8a.2), transmission (Fig 8a.3 and 8a.4) and ICU use (Fig 8a.5). Yet, in the scenario where all churches would be effectively closed, we found a reduction of around 10% in the number of deaths (median of 954, (885, 925, 980, 1,030, Fig 8b.1 and 8b.2) and increased number of transmissions (Fig 8b.3 and 8b.4) and ICU demand (Fig 8b.5). As in the work-related analysis, we found that a delayed but effective decree would have been a better option than the strategy that was actually used. As our simulation suggests, we can estimate that implementing the laws effectively, even with a delay, would spare nearly 100 lives (Fig 8c.1 and 8c.2).

## What if no intervention was ever applied?

In the last scenario, we asked what would be the outcome if there was no intervention from the government but the regulation of the use of face masks. The simulations show that this scenario would lead to high levels of daily deaths with a rapid increase from April to July, reaching the peak on July 4th (median of 73 daily deaths) (Fig 9A). After this month, there is a slow decrease in the following months. The accumulated deaths (median 7,966 (7,443, 7,729, 8,248, 8,647), Fig 9B) show an enormous increase of 6,894 deaths when compared with the baseline scenario (Fig 5). A similar effect is observed in the number of infections (Fig 9C, 9D and 9F) and in the ICU demand (Fig 9F).

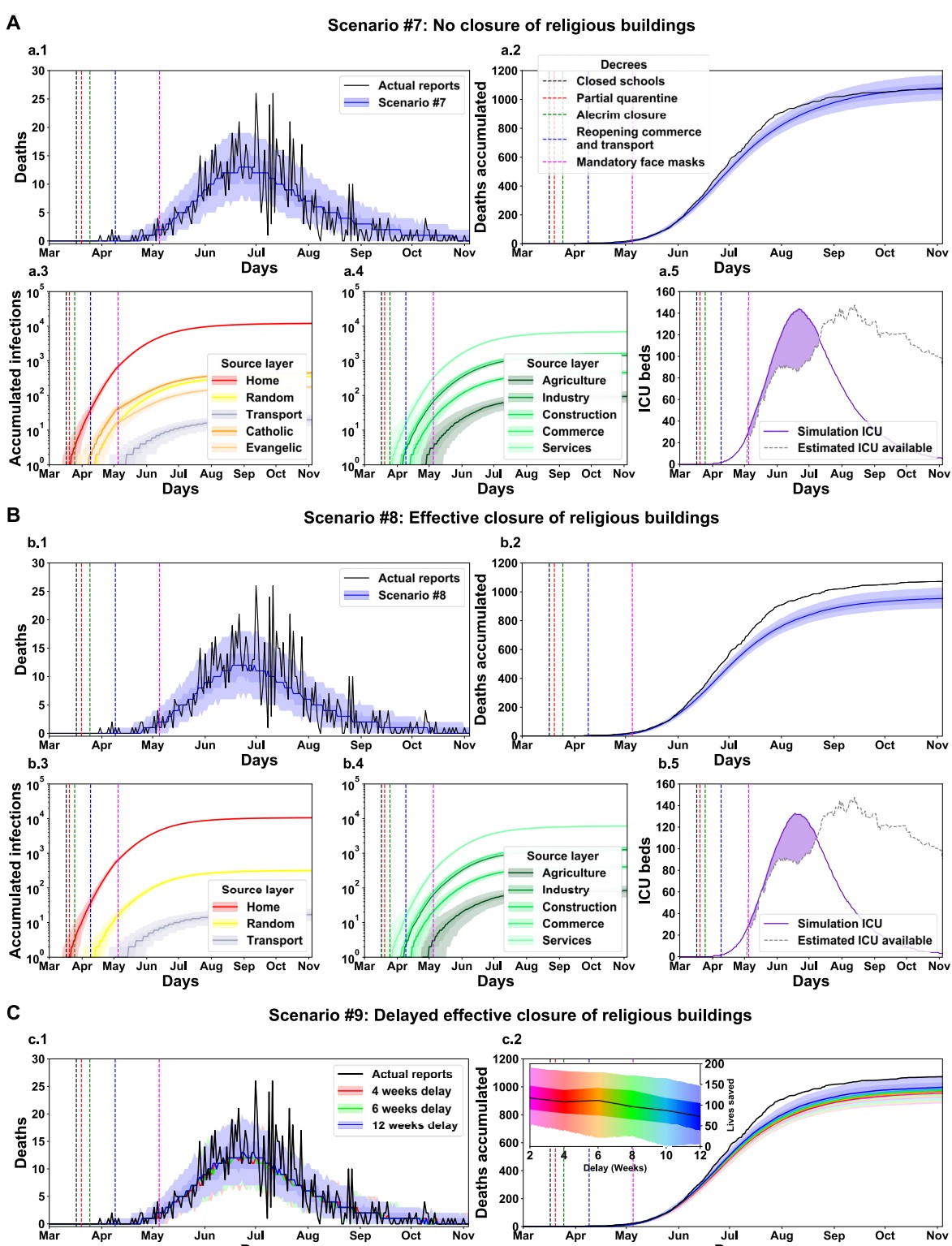

**Fig 8. Simulation of different scenarios reveals a moderate and sub-optimal impact of the worship venues closure decree on March 25th, 2020.** Results for a baseline scenario altered for the condition that worship venues were never closed (#7, A), closed effectively (#8, B) and closed effectively with a delay from the original decree's publication date (#9, C). The total number of lives saved in each scenario is the difference between the reported number of deaths and the simulation outcome. For scenario #9, the delay in weeks is colour-coded, and the additional number of lives saved is shown in an insert. See Fig 5 for panel description.

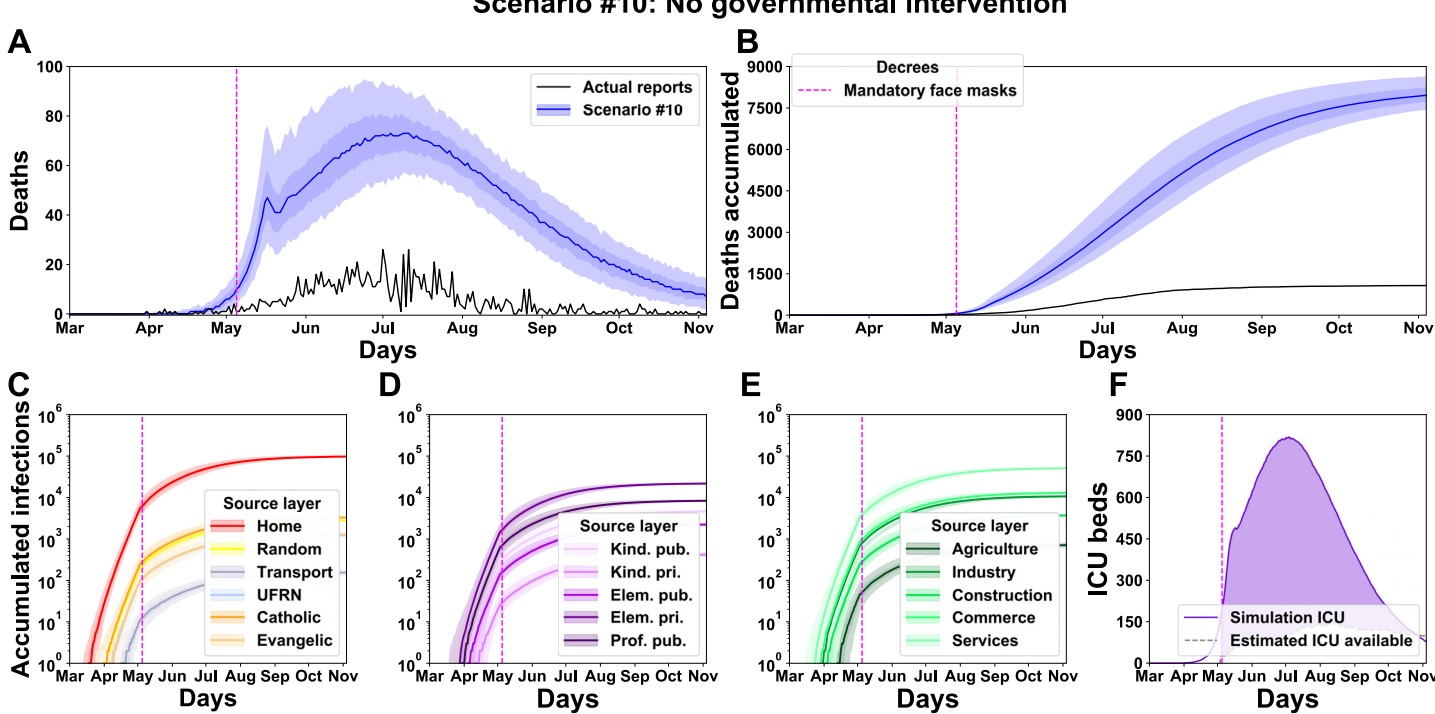

**Fig 9. Simulation of a scenario without any decree implies a catastrophic scenario.** Results for a baseline scenario altered for the condition that no decree was ever published. Panels from A to F, as in Fig 5.

## Discussion

This study presented an agent-based epidemiological model designed to investigate the impact of governmental non-pharmaceutical interventions in the first SARS-CoV-2 wave in the City of Natal (February 26th to November 4th of 2020). The model could support simulations with a 1:1 agents/citizens ratio and used detailed data on (1) health data from daily epidemiological reports, (2) demographic data of Natal city, (3) estimates of external cases arising from airport, bus station and highway flows; and (4) government decrees to confront the pandemic. The model health states extended the SIR model with additional states implemented to represent the complexity of the Covid-19 pathology. We implemented a complex network to emulate the contact among agents. The networks subdivide into layers and sub-layers to better characterise the social interactions such as schools, workplaces, churches, transport, and others, resulting in each network having its own specificities. The simulations successfully reproduced the observed curve of deaths and allowed an estimation of the number of infections and hospitalisations (Fig 5).

Altogether, the results of the simulations support that the actions of state and local governments effectively reduced the loss of citizen lives in the first SARS-CoV-2 wave in the City of Natal. Although the outcome was not optimal—as more rigid interventions could have saved more lives—all interventions seem to have had a positive impact. The model can support the conjecture of hypothetical scenarios with different government interventions. We simulated three main scenarios focused on: (1) school layers, (2) workplace layers, and (3) religious layers. Table 5 summarises the impact of the interactions among the three main layer groups (schools, workplaces and religions) within each simulated scenario. We identified that the work layers operate as a hub of infections. Although the scenario without a decree for closing

**Table 5. Summary of simulation results.**

|  | Scenarios | Layers | | | Outcomes | | |
| --- | --- | --- | --- | --- | --- | --- | --- |
|  |  | Schools | Work | Religion | Impact | Deaths | Difference of deaths |
| Fig 5 | Baseline | X | - | - |  | 1073 |  |
| Fig 6A | #1 Schools never closed | + | - | - | >> | 6332 | 5259 |
| Fig 7A | #4 Businesses never closed | X | + | - | > | 1240 | 167 |
| Fig 7B | #5 Effective closure of businesses | X | X | - | << | 317 | -756 |
| Fig 8A | #7 No closure of religious buildings | X | - | + | > | 1081 | 8 |
| Fig 8B | #8 Effective closure of religious buildings | X | - | X | < | 954 | -119 |
| Fig 9 | #10 No governmental intervention | + | + | + | >>> | 7966 | 6893 |

Labels: − Low interaction; + High interaction; X No interaction; > Increase; < Decrease; >> High increase; << High decrease; >>> Higher increase.

workspaces (scenario Fig 7A) resulted in a low increase of deaths, the workspace layers can support a highly infectious dynamic when operating combined with another group of layers, even when the strength of the interaction is low. This effect surges in scenarios without the decrees to close schools (Fig 6A) or churches (Fig 8A). When there is no interaction in these layers, the number of deaths decreases (Fig 7B—Close workplaces effective decree). As the main finding of the simulations with the absence of a Schools closing decree, the model shows a high increase of deaths, around 591.6% (Fig 6A). Nevertheless, the principal finding for workplaces and religious layers was the effective implementation of the decrees. With the absence of activities in the work layers, the number of death decreased by 70.43% (Fig 7B). The closure of religious layers would reduce the ratio of death by 11.01% (Fig 8B). The absence of intervention would result in a catastrophic scenario of 7966 deaths, which corresponds to around 0.912% of the entire Natal city population (Fig 9). We can use the model to quantify the number of infected by layers/sub-layers, showing that, typically, the agents get infected primarily in the Home layer. Public education was the most contagious among the School layers' simulated scenarios. The Service sub-layer is the most infectious work activity. The most affected religion was Catholicism. These observations strongly support the need for active governmental action in the face of a pandemic such as SARS-CoV-2 ranging from a wide diversity of activities and economic areas.

The modelling approach used in this study—based on agents—allows insights into internal aspects of the pandemic evolution that would be difficult—but not impossible—in a compartment-based paradigm. One could model the multiple transmission networks as variable channels between compartments and separate the compartments based on multiple social and health criteria. However, the rapid explosion of the number of compartments and networks would preclude the main advantage of compartment-based models: simplicity. Agent-based models allow a simple inclusion of many aspects of the phenomena with much simpler steps. Still, it is essential to remember that the simplicity of including variables could also result in models that are hard to associate with reality and possibly misleading. For instance, we do not explicitly model the adherence levels to the government policies or whether they vary with the agent's attributes, such as sex, age, region, occupation, and stratum. There is evidence that adherence levels varied according to various circumstances, including political views [54]. But, also, there are significant indications of widespread commitment to the policies, as indicated by the overall reduction of mobility reported by local telecommunication companies [55] and the result of online inquiries [56]. Still, we did not find a reliable dataset that allowed us to assess the population's adherence to the guidelines quantitatively; thus, we did not explicitly

model this aspect of the implementation of government policies. Nevertheless, the methodology would be relatively simple to adapt if such data is available in a different city.

Also, due to the lack of data and some level of abstraction necessary for computation, we advise the reader to interpret the results considering its methodological limitations. For example, the baseline scenario considered in this study slightly underestimated the cumulative death curve for a short period after the peak death rate (Fig 5). Although the high R-squared value suggests that we came across a representative fit, the short underestimation could also indicate that our search methodology found a local maximum or that the current level of abstraction of the model is insufficient for a perfect fit to baseline. Still, an alternative scenario with a potentially better fit would probably render similar results considering the relative impact of the different transmission layers (Fig 4).

This study did not analyse the second SARS-CoV-2 wave in the City of Natal. The second wave was longer, deadlier, and more complex, resulting in over 2700 deaths by December 2021. We did not consider the second wave because most of the governmental decrees—all but the closure of public schools—became ineffective as the population showed a decreased respect for the measures. As an illustrative example, the Brazilian government decided to proceed with the city elections on November 15th. Traditionally, politicians organise public events for their supporters before election day, and although the authorities tried to limit them, these events still occurred [57]. Also, travelling resumed at the end of the year for the two major holidays for the Brazilian population: Christmas and Reveillon [58].

Additionally, there were local reports of several clandestine Carnaval events that occurred on February 2021 [59]. Further, vaccines became progressively widely available in Brazil during the second wave, and, likely, the impact of delayed distribution in the development of the epidemy was more determinant than mobility restrictions. The vaccination in Brazil started in January 2021, prioritising health professionals, older people (60+ years), and indigenous [60]. Although Brazil has a comprehensive and free health system that supports everyone, the population immunisation rhythm was slow due to mismanagement in acquiring and distributing vaccines [61]. The model can be adapted to study the effect of vaccination in the pandemic, but we leave it as a possible follow-up.

## Acknowledgments

This research was supported by the High Performance Computing Center at UFRN (NPAD/ UFRN). We thank the State Agency for Púplic Health of Rio Grande do Norte (Secretaria de Estado de Saúde Pública do Rio Grande do Norte, SESAP) for support in data acquisition.

## Author Contributions

**Conceptualization:** Paulo Henrique Lopes, Liam Wellacott, Leandro de Almeida, Lourdes Milagros Mendoza Villavicencio, Luciana Lima, Michael Lones, José-Dias do Nascimento, Jr., Patricia A. Vargas, Renan Cipriano Moioli, Wilfredo Blanco Figuerola, César Rennó-Costa.

**Data curation:** Paulo Henrique Lopes, Liam Wellacott, Leandro de Almeida, Lourdes Milagros Mendoza Villavicencio, André Luiz de Lucena Moreira, Rislene Katia Ramos de Sousa, Priscila de Souza Silva, Luciana Lima, Renan Cipriano Moioli, Wilfredo Blanco Figuerola, César Rennó-Costa.

**Formal analysis:** Paulo Henrique Lopes, Liam Wellacott, Leandro de Almeida, Lourdes Milagros Mendoza Villavicencio, André Luiz de Lucena Moreira, Rislene Katia Ramos de Sousa,

Priscila de Souza Silva, Michael Lones, Renan Cipriano Moioli, Wilfredo Blanco Figuerola, César Rennó-Costa.

**Funding acquisition:** Michael Lones, Patricia A. Vargas, Renan Cipriano Moioli, César Rennó-Costa.

**Investigation:** Paulo Henrique Lopes, Liam Wellacott, Leandro de Almeida, Lourdes Milagros Mendoza Villavicencio, Rislene Katia Ramos de Sousa, Priscila de Souza Silva, Luciana Lima, Michael Lones, José-Dias do Nascimento, Jr., Patricia A. Vargas, Renan Cipriano Moioli, Wilfredo Blanco Figuerola, César Rennó-Costa.

**Methodology:** Paulo Henrique Lopes, Liam Wellacott, Leandro de Almeida, Lourdes Milagros Mendoza Villavicencio, Rislene Katia Ramos de Sousa, Priscila de Souza Silva, Luciana Lima, Michael Lones, José-Dias do Nascimento, Jr., Renan Cipriano Moioli, Wilfredo Blanco Figuerola, César Rennó-Costa.

**Project administration:** Liam Wellacott, Patricia A. Vargas, Renan Cipriano Moioli, César Rennó-Costa.

**Resources:** Paulo Henrique Lopes, Liam Wellacott, Leandro de Almeida, Lourdes Milagros Mendoza Villavicencio, André Luiz de Lucena Moreira, Dhiego Souto Andrade, Alyson Matheus de Carvalho Souza, Rislene Katia Ramos de Sousa, Renan Cipriano Moioli, Wilfredo Blanco Figuerola, César Rennó-Costa.

**Software:** Paulo Henrique Lopes, Liam Wellacott, Leandro de Almeida, André Luiz de Lucena Moreira, Alyson Matheus de Carvalho Souza, Michael Lones, Renan Cipriano Moioli, Wilfredo Blanco Figuerola, César Rennó-Costa.

**Supervision:** Paulo Henrique Lopes, Luciana Lima, Michael Lones, José-Dias do Nascimento, Jr., Patricia A. Vargas, Renan Cipriano Moioli, Wilfredo Blanco Figuerola, César Rennó-Costa.

**Validation:** Paulo Henrique Lopes, Liam Wellacott, Leandro de Almeida, Dhiego Souto Andrade, Alyson Matheus de Carvalho Souza, Luciana Lima, Michael Lones, Renan Cipriano Moioli, Wilfredo Blanco Figuerola, César Rennó-Costa.

**Visualization:** Paulo Henrique Lopes, Liam Wellacott, Leandro de Almeida, Dhiego Souto Andrade, Renan Cipriano Moioli, Wilfredo Blanco Figuerola, César Rennó-Costa.

**Writing – original draft:** Paulo Henrique Lopes, Renan Cipriano Moioli, César Rennó-Costa.

**Writing – review & editing:** Paulo Henrique Lopes, Leandro de Almeida, Dhiego Souto Andrade, Alyson Matheus de Carvalho Souza, Luciana Lima, Michael Lones, José-Dias do Nascimento, Jr., Patricia A. Vargas, Renan Cipriano Moioli, Wilfredo Blanco Figuerola, César Rennó-Costa.

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
