## [Editor Report · Decision Letter 0]

20 May 2022

PGPH-D-22-00736

Measuring the impact of nonpharmaceutical interventions on the SARS-CoV-2 pandemic at a city level: An agent-based computational modeling study of the City of Natal.

Dear Dr. Rennó-Costa,

Thank you for submitting your manuscript to PLOS Global Public Health. After careful consideration, we feel that it has merit but does not fully meet PLOS Global Public Health’s publication criteria as it currently stands. Therefore, we invite you to submit a revised version of the manuscript that addresses the points raised during the review process.

Please submit your revised manuscript by . If you will need more time than this to complete your revisions, please reply to this message or contact the journal office at globalpubhealth@plos.org. Please include the following items when submitting your revised manuscript:

We look forward to receiving your revised manuscript.

Kind regards,

Javier H Eslava-Schmalbach, M.D., Ph.D., MSc

Academic Editor

Journal Requirements:

State what role the funders took in the study. If the funders had no role in your study, please state: “The funders had no role in study design, data collection and analysis, decision to publish, or preparation of the manuscript.”

2. Please send a completed 'Competing Interests' statement, including any COIs declared by your co-authors. If you have no competing interests to declare, please state “The authors have declared that no competing interests exist”. Otherwise please declare all competing interests beginning with the statement "I have read the journal's policy and the authors of this manuscript have the following competing interests:

3. Please ensure that you provide a single, cohesive .tex source file for your LaTeX revision. You may upload this file as the item type 'LaTeX Source File.' As stated in the PLOS template, your references should be included in your .tex file (not submitted separately as .bib or .bbl). Please also ensure that you are making any formatting changes to both your .tex file and the PDF of your manuscript. If you have any questions, please contact Latex@plos.org. You can find our LaTeX guidelines here: 

https://journals.plos.org/globalpublichealth/s/latex "

4. Please provide separate figure files in .tif or .eps format and remove any figures embedded in your manuscript file. Please also ensure that all files are under our size limit of 10MB. If you are using LaTeX, you do not need to remove embedded figures.

5. Please provide an Author Summary. This should appear in your manuscript between the Abstract (if applicable) and the Introduction, and should be 150–200 words long. The aim should be to make your findings accessible to a wide audience that includes both scientists and non-scientists. Sample summaries can be found on our website under Submission Guidelines: https://journals.plos.org/globalpublichealth/s/submission-guidelines#loc-parts-of-a-submission

Alternative link: http://journals.plos.org/ploscompbiol/s/submission-guidelines#loc-author-summary

6. We notice that your supplementary tables are included in the manuscript file. Please remove them and upload them with the file type 'Supporting Information'. Please ensure that each Supporting Information file has a legend listed in the manuscript after the references list.

Additional Editor Comments (if provided):

Dear authors

We appreciate the submission of your paper to our journal. Even when the way you submitted the manuscript let us to know how it is going to be its layout versión, it is not the appropriate one, that is suggested in the "Instructions for Authors". It is important to follow this Instructions, given that it makes easier the the reviewers' work. At the end of your submission you could add, your "layout version".
---

## [Decision Letter · Decision Letter 1]

20 Jul 2022

PGPH-D-22-00736R1

Measuring the impact of nonpharmaceutical interventions on the SARS-CoV-2 pandemic at a city level: An agent-based computational modeling study of the City of Natal.

Dear Dr. Rennó-Costa,

Thank you for submitting your manuscript to PLOS Global Public Health. After careful consideration, we feel that it has merit but does not fully meet PLOS Global Public Health’s publication criteria as it currently stands. Therefore, we invite you to submit a revised version of the manuscript that addresses the points raised during the review process.

We look forward to receiving your revised manuscript.

Kind regards,

Javier H Eslava-Schmalbach, M.D., Ph.D., MSc

Academic Editor

Journal Requirements:

Additional Editor Comments (if provided):

Dear Authors:

We have finally received the peer-reviewers' comments. Please comment, include, accept all and each one of them. Additionally, please comment these considerations, that should be explained within the text or in limitations:

1. How did you deal with differences between government policies and adherence to them, from all the actors involved, considering that they also had different behaviors during the pandemic that affected this adherence. Real data were affected by these behaviors, and in some cases they could be different according to their places of residence, age and socioeconomic status. Apparently, simulation did not consider this.

2. You mentioned that the "incubated state follows a log-normal distribution". It is not understandable why did you include estimated means and standard deviations of death values, assuming that they were normal. In fact, in some of the figures, means and standard deviations are far away from real data. How do you interpret these differences in Figure 5A, 5B, 6B, 8A.

Reviewers' comments:

Reviewer's Responses to Questions

**Comments to the Author**

1. If the authors have adequately addressed your comments raised in a previous round of review and you feel that this manuscript is now acceptable for publication, you may indicate that here to bypass the “Comments to the Author” section, enter your conflict of interest statement in the “Confidential to Editor” section, and submit your "Accept" recommendation.

Reviewer #1: (No Response)

Reviewer #2: All comments have been addressed

2. Does this manuscript meet PLOS Global Public Health’s publication criteria? Is the manuscript technically sound, and do the data support the conclusions? The manuscript must describe methodologically and ethically rigorous research with conclusions that are appropriately drawn based on the data presented.

Reviewer #1: No

Reviewer #2: Yes

3. Has the statistical analysis been performed appropriately and rigorously?

Reviewer #1: N/A

Reviewer #2: Yes

4. Have the authors made all data underlying the findings in their manuscript fully available (please refer to the Data Availability Statement at the start of the manuscript PDF file)?

Reviewer #1: Yes

Reviewer #2: Yes

5. Is the manuscript presented in an intelligible fashion and written in standard English?

Reviewer #1: Yes

Reviewer #2: Yes

6. Review Comments to the Author

Reviewer #1: The manuscript describes use of an agent-based model to assess intervention strategies used during the COVID-19 pandemic. I have two major concerns:

- The authors do not describe how they optimized the model to fit the data used for validation. They need to explain which parameters were adjusted to fit the data and what algorithm was used. The baseline curve they ended up with underestimates the cumulative deaths for a period of time. It's also not clear if the government decrees were incorporated into the fit by changing parameters at the time the decrees went into effect. Since the baseline parameters help determine the relative importance of different interactions in spreading the virus, having incorrect baseline values will lead to incorrect predictions of the effect of different interventions.

- A global sensitivity analysis might be more useful here to assess which classes or sub-classes contribute most to the number of fatalities.

Minor comments:

- The figures are of low resolution and need to be improved. It's also not clear what the inset graph on the final graphs of figures 6, 7, and 8 is supposed to represent.

- Line 8: The 18 million deaths mentioned here is rather high compared to the 6 million or so officially recorded COVID deaths to date. The authors should make clear that the 18 million figure is based on excess mortality estimates and that not all these deaths are directly attributable to COVID.

- The description of adding external infections is not entirely clear. Are these new agents added to the model or are the appropriate number of new infections randomly assigned to the existing population?

- Line 501: One of the two dates (June 19th or June 16th) is incorrect, since the curve cannot reach the peak on the 19th and "remain there" to three days before it reaches the peak.

- While the English is generally clear, there are places where incorrect terminology or wording is used. The manuscript should be checked by a native English speaker.

Reviewer #2: Overall, this is an interesting application study. The evaluation of the effect of governmental measures in order to estimate their effects and to help to prepare for future events is really important. While the modeling strategy seems similar to other models in the literature (which makes its novelty weaker), I see the results presented interesting. It would make a reacher contribution to put into perspective the reason why one would like to extend the SIR model and what is the effect of this extra complexity in the quality of the modeling and also to put it into the context of other similar studies. Right now the paper presents no discussion or justification for this.

Additionally, some points were not clear to me. In particular, on page 11 the paper introduces equation 1 that has this variable Pcontamination (which the equation uses a small p for pcontamination, while the text uses capital P). On page 17, ,subsection Decrees, the phrase "We did not set the Pcontamination value to zero because some establishments continued to work" seems strange to me. Since, if on is allowed to set Pcontamination to a low value, this would automatically say that the decrees were effective, which I believe it is something that one wants to validate and fit from the data. So I believe this has to be made clear in the paper.

Other criticism concerning the paper is related to its organization: in many points the paper is verbose and repeats itself in multiple occasions, there is in many places text . The figures are now well organized, being convoluted with too much information and hard to interpret at times. I would suggest that these figures should be reworked especially figure 2, from which it is hard to extract information from and figures 6 and 7. Finally, there are many numbers concerning the data from the city and its different areas that I believe could be condensed in a table, instead of being scattered though the paper.

7. PLOS authors have the option to publish the peer review history of their article (what does this mean?). If published, this will include your full peer review and any attached files.

**Do you want your identity to be public for this peer review?** For information about this choice, including consent withdrawal, please see our Privacy Policy.

Reviewer #1: No

Reviewer #2: No

---

## [Editor Report · Decision Letter 2]

26 Sep 2022

Measuring the impact of nonpharmaceutical interventions on the SARS-CoV-2 pandemic at a city level: An agent-based computational modelling study of the City of Natal.

PGPH-D-22-00736R2

Dear Prof Rennó-Costa,

We are pleased to inform you that your manuscript 'Measuring the impact of nonpharmaceutical interventions on the SARS-CoV-2 pandemic at a city level: An agent-based computational modelling study of the City of Natal.' has been provisionally accepted for publication in PLOS Global Public Health.

Best regards,

Javier H Eslava-Schmalbach, M.D., Ph.D., MSc

Academic Editor